# Sex-Differences in the Pattern of Comorbidities, Functional Independence, and Mortality in Elderly Inpatients: Evidence from the RePoSI Register

**DOI:** 10.3390/jcm8010081

**Published:** 2019-01-12

**Authors:** Salvatore Corrao, Christiano Argano, Giuseppe Natoli, Alessandro Nobili, Gino Roberto Corazza, Pier Mannuccio Mannucci, Francesco Perticone

**Affiliations:** 1Department of Internal Medicine, National Relevance and High Specialization Hospital Trust ARNAS Civico, Di Cristina, Benfratelli, Palermo 90127, Italy; chargano@yahoo.it (C.A.); peppenatoli@gmail.com (G.N.); 2Materno Infantile, Medicina Interna e Specialistica di Eccellenza “G. D’Alessandro”, PROMISE, Dipartimento di Promozione della Salute, Università di Palermo, Palermo 90133, Italy; 3Department of Organizational, Clinical, and Translational Research, I.E.ME.S.T., Palermo 90139, Italy; 4Department of Neuroscience, IRCCS, Istituto di Ricerche Farmacologiche Mario Negri, Milan 20156, Italy; alessandro.nobili@marionegri.it; 5Department of Internal Medicine, University of Pavia and San Matteo Hospital, Pavia 27100, Italy; gr.corazza@smatteo.pv.it; 6Scientific Direction, IRCCS Foundation Maggiore Policlinico Hospital, Milan 20122, Italy; piermannuccio.mannucci@policlinico.mi.it; 7Department of Medical and Surgical Sciences, University Magna Graecia of Catanzaro, Catanzaro 88100, Italy; perticone@unicz.it

**Keywords:** elderly, sex profiles, disease distribution, in-hospital mortality, 3-month mortality, 1-year mortality

## Abstract

Background: The RePoSi study has provided data on comorbidities, polypharmacy, and sex dimorphism in hospitalised elderly patients. Methods: We retrospectively analysed data collected from the 2010, 2012, 2014, and 2016 data sets of the RePoSi register. The aim of this study was to explore the sex-differences and to validate the multivariate model in the entire dataset with an expanded follow-up at 1 year. Results: Among 4714 patients, 51% were women and 49% were men. The disease distribution showed that diabetes, coronary artery disease, chronic obstructive pulmonary disease, chronic kidney disease, and malignancy were more frequent in men but that hypertension, anaemia, osteoarthritis, depression, and diverticulitis disease were more common in women. Severity and comorbidity indexes according to the Cumulative Illness Rating Scale (CIRS-s and CIRS-c) were higher in men, while cognitive impairment, mood disorders, and disability in daily life measured by the Barthel Index (BI) were worse in women. In the multivariate analysis, BI, CIRS, and malignancy significantly increased the risk of death in men at the 1-year follow-up, while age was independently associated with mortality in women. Conclusions: Our study highlighted the relevance and the validity of our previous predictive model in the identification of sex dimorphism in hospitalised elderly patients underscoring the need of sex-personalised health-care.

## 1. Introduction

In Europe by 2060, people aged 65 and older will become 28% of the whole population, whereas the oldest old will be 12% as numerous as young people [1]. It is quite clear that ageing is characteried by multiple chronic diseases. Some studies have evaluated the differences between men and women in the clinical manifestations of the most frequent diseases [2] and drug prescriptions [3]. We previously published data from the RePoSi register that evidenced clinical, epidemiological sex differences of the elderly population hospitalised in internal medicine and geriatric Italian wards [4]. In particular, we highlighted that sex difference, comorbidity profiles and age, male sex, low Barthel index, in addition to malignancy, are strong predictors of mortality at a three-month follow-up. This first analysis was based only on the 2010 RePoSi data set, from a sample size of 1380 patients enrolled by 66 wards with a limited short outcome. Since the RePoSi is a prospective and multicentre study that continues to collect data, at the moment, a bigger sample size of 4714 patients enrolled by 101 wards is available as well as 1-year follow-up. Given this background, we aimed to verify our preliminary findings in a large dataset expanding the predictive model to a 1-year follow-up and including a specific validation of the multivariate model both for men and for women.

## 2. Material and Methods

The RePoSi is a collaborative and independent register of the Italian Society of Internal Medicine (SIMI), the Mario Negri Institute for Pharmacological Research, and the IRCCS Foundation Maggiore Policlinico Hospital. The design of RePoSi was previously described in detail [5]. Data from the 2008, 2010, 2012, 2014, and 2016 recruitments and different clinically relevant variables are currently available. However, the 2008 cohort had no post-discharge follow-up. For this reason, the 2008 dataset was not considered in all the analyses. The sampling of each cohort was as follow: 1380 patients in 2010, 1323 in 2012, 1212 in 2014, and 799 in 2016. This observational study was carried out following the Strengthening the Reporting of Observational Studies in Epidemiology (STROBE) Statement standard reporting [6].

The following clinical characteristics were evaluated: disease distribution at hospital admission (classification was based on the International Classification of Diseases, Ninth Revision); cognitive status according to the Short Blessed Test (SBT); [7] mood disorders using the Geriatric Depression Scale (GDS) score [8]; performance in basic activities of daily living (measured by means of the Barthel Index and [9] classified as mild disability (BI 75–90), moderate disability (BI 50–74), severe disability (BI 25–49), and completely dependent (BI 0–24)); comorbidity and severity indexes (according to the Cumulative Illness Rating Scale (CIRS-c and CIRS-s)) [10]; kidney function by means of the eGFR (calculated by the Chronic Kidney Disease Epidemiology Collaboration formula) [11]; and in-hospital, 3-month, and 12-month mortality rates. Moreover, age; systolic and diastolic blood pressure; creatinine; eGFR; haemoglobin; fasting glucose; cholesterol; BMI; length of hospital stay; in-hospital, 3-month, and 1-year mortalities; number of drugs on hospital admission; discharge; and at 3-month and 1-year follow-ups were also evaluated. The association between the variables and mortality (in-hospital and at 3-month and 1-year follow-ups) was analysed. The study was approved by the Ethics Committee of RePoSI and by all local committees.

### Statistical Analysis

Data were reported as percentages for categorical variables and as means (95% confidence intervals) for quantitative variables. The exact Fisher test for the contingency tables and the z test for proportions were used for comparison between the groups. The comparison of the quantitative variables was made using the non-parametric Mann–Whitney U test. A multivariate logistic analysis was used to explore the relationship between the variables and outcomes. Odds ratios (ORs) and 95% confidence intervals (95% CIs) were computed. The variables were chosen according to the Hosmer–Lemeshow methodology [12]: after the univariate analysis, only variables with a *p* < 0.20 were included in the final model; then, through a backward process, variables were excluded until a significance level of *p* < 0.05 was reached for each variable. Three multivariate models were performed for each outcome period (in-hospital, 3-month, and 1-year). All multivariate analyses were performed using different cohort samples as clusters. The new multivariate models were tested in the whole population for both men and women according to the cluster categorisation that was incorporated in every analysis (including confidence interval computation in cumulated cohort data). The different cohort samples were used as clusters.

A two-tailed *p* < 0.05 was considered statistically significant. Stata (StataCorp. 2016. Stata Statistical Software: Release 14.1, College Station, TX, USA: StataCorp LP) was used for database management and analysis. 

## 3. Results

A total of 4714 hospitalised patients aged 65 years or older were eligible for the analysis. Both sexes were equally represented. Women were older than men. Noteworthily, there was a greater percentage of women in the 80 years and older age group (Figure 1). 

The analyses of the clinical characteristics of the whole population according to sex (Table 1) showed that eGFR, creatinine, haemoglobin; and the number of drugs at hospital admission were higher in men but that age and serum cholesterol were more elevated in women.

Women showed a higher disability from the Barthel Index and geriatric depression scale. Notably, the analysis of disability according to age classes showed that women between 70 and 90 years old had a Barthel index score significantly lower than men (Figure 2). 

Men had a higher in-hospital, 3-month, and 1-year mortalities and comorbidity and severity index. In the 70–90 year age groups, men had a severity index significantly higher than women (Figure 3).

The trend for length of the hospital stay is comparable. The disease distribution showed that malignancy, diabetes, coronary artery disease, chronic kidney disease, and chronic obstructive pulmonary disease were more frequent in men, whereas hypertension, osteoarthritis, anaemia, depression, and diverticulitis disease occurred more often in women (Figure 4). 

A multivariate model of all data available was done (Table 2). 

Sex, age, and the Barthel index were independently associated with in-hospital, 3-month, and 1-year mortalities. Malignancy was a strong predictor of mortality at the 3-month and 1-year follow-ups as well. Since sex was a variable independently associated with mortality, two sex-specific multivariate analyses have been done. Age and the Barthel Index were independently associated with in-hospital mortality both in men and in women. In addition, CIRS was a predictor of mortality in men. At the 3-month follow-up age, the Barthel index and malignancy were independently associated with mortality in both men and women. At the 1-year follow-up Barthel index, CIRS and malignancy were predictors of death in men while only age increased the risk of death in women (Figure 5 and Figure 6). 

## 4. Discussion

In our previous study, we showed the existence of sex differences regarding clinical characteristics and outcomes [4]. This second analysis that included data of a more representative sample size confirmed our predictive model about sex-dimorphism in hospitalised elderly patients. Women were older than men in agreement with previous epidemiological data [13]. Women still had a worse GDS score than men. It is important to highlight that according to the Nurses’ Health Study, women with clinical depression were at an increased risk of stroke [14] and cardiovascular events [15]. Our findings, obtained using the Short Blessed Test, confirmed that cognitive impairment was more evident in women. According to the Framingham study, the age-specific risk of Alzheimer’s disease was almost twofold higher in women than men [16]. Lin et al. showed that women with mild cognitive impairment had a higher rate of decline in cognitive performance and functional status than men [17]. Different studies have tried to explore the reasons for this sexual difference. The ε4 allele of the apolipoprotein ε (APOE) was associated with an earlier onset of Alzheimer’s disease [18]. Some authors showed that women with one or more APOE-epsilon4 allele were more likely to develop Alzheimer’s disease in comparison with men [19,20]. Moreover, Fukumoto et al. [21] showed that the brain-derived neurotrophic factor (BDNF) Met66 allele, which reduces the transport of BDNF, is linked to an increased risk of Alzheimer’s disease in women since estrogen plays a main role in the expression of BDNF [22]. Cognitive impairment and depression may lead to the development of limitations in activities of daily living, contributing to a progressive functional disability [23]. Our analysis confirmed that women had a worse Barthel Index in comparison with men, emphasizing, again, the link among ageing, psychological attitude, mental status, and activities of daily living. In addition, the disease distribution highlights a male profile more prone to be affected by chronic obstructive pulmonary disease (COPD), diabetes mellitus, coronary artery diseases, and malignancies, resulting in a higher number of medications taken at hospital admission, hospital discharge, and 3-month and 1-year follow-ups and determining a higher in-hospital, 3-month, and 1-year mortalities. Our findings are in agreement with NHANES data that showed a higher prevalence of COPD in men in comparison with women [24]. It is worth outlining that gender phenotypic differences in COPD expression have been highlighted: Women have a lower prevalence of emphysema than men while bronchitis is more frequent in women. Dransfield et al. [25], using computed tomography and histology, have found similar results showing that men had more severe emphysema in comparison with women that had significantly thicker airway walls and smaller lumens.

On the other hand women were affected more frequently by hypertension, dementia, depression, osteoarthritis, rheumatic diseases, anemia, and diverticular disease that are responsible for further impairment in daily life activities. These data are consistent with previous studies that showed similar prevalences of such conditions [26]. Regarding this, the main role of sex hormones must be outlined. Estrogens contribute to the lower prevalence of hypertension in women than men comparable for age [27]. In addition, estrogens along with prolactin are associated with female predisposition to autoimmune and rheumatic diseases [28,29], and sex hormones are involved in the upregulation of the expression of higher adiponectin levels that reduces the risk of type 2 diabetes in women [30]. Moreover, female sex hormones exert a protective role in the premenopausal years [31], determining a development of cardiovascular disease 10 years later than men [32]. Recently we outlined the considerable role of age, sex, and the severity index that overcome cardio-metabolic comorbidities as powerful predictors of mortality in elderly subjects [33]. In this analysis, age and the Barthel Index were the strongest predictors of in-hospital, 3-month, and 1-year mortalities. In this sense, a recent study showed that the Barthel index was independently associated with mortality in the oldest old patients hospitalised with pneumonia [34]. On the men side, the severity index was independently associated with in-hospital and 1-year mortalities, whereas malignancy was independently associated with 3-month and 1-year mortalities. Once again, our data showed that the male sex was more affected than women with regards to cumulative illness burden. In women, malignancy was independently associated with 3-month mortality. Similar to the 2010 analysis, systolic blood pressure had a protective role regarding in-hospital and 3-month mortalities and is consistent with epidemiological studies that demonstrated an inverse association between higher blood pressure and mortality in the oldest old patients [35,36]. We believe that this could be the benchmark in terms of variables involved in the prognosis of hospitalised elderly patients. This analysis strengthens the findings of our previous study, expanding our knowledge to a 1-year period, making it more robust. 

The most important strength of our analysis is the expansion and update of our previous model in the frame of a bigger sample size and great statistical consistency. Odds ratio and confidence intervals were defined by more robust statistical methods and by a larger sample size. 

## 5. Conclusions

In conclusion, an evident sex dimorphism in the clinical and morbidity characteristics is present in the elderly population hospitalised in internal medicine and geriatric wards: Women were older than men and more often affected by chronic conditions reducing their daily life activities, while men did show more multiple chronic conditions with higher short-term and long-term mortality following hospitalisation. The identification of these differences could represent a powerful tool to promote sex-personalised care in elderly inpatients. Our data could be used to rethink and re-engineering a structural model of the relationship between hospital and health care services to avoid and reduce rehospitalisation burdens.

## Figures and Tables

**Figure 1 jcm-08-00081-f001:**
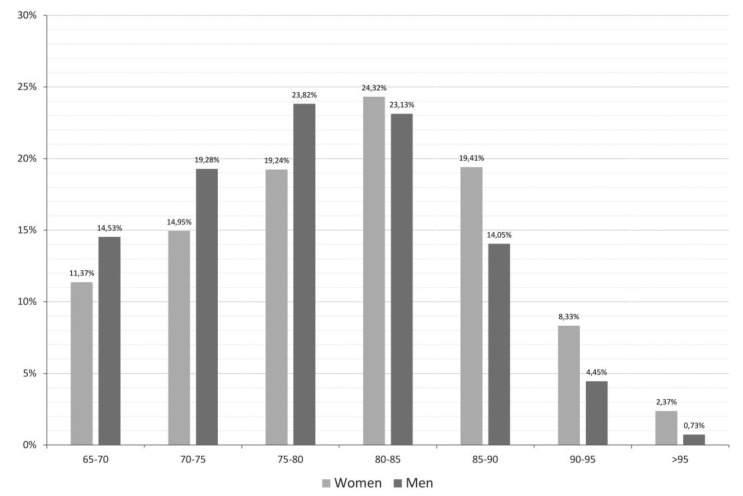
Frequency distribution according to sex and age classes.

**Figure 2 jcm-08-00081-f002:**
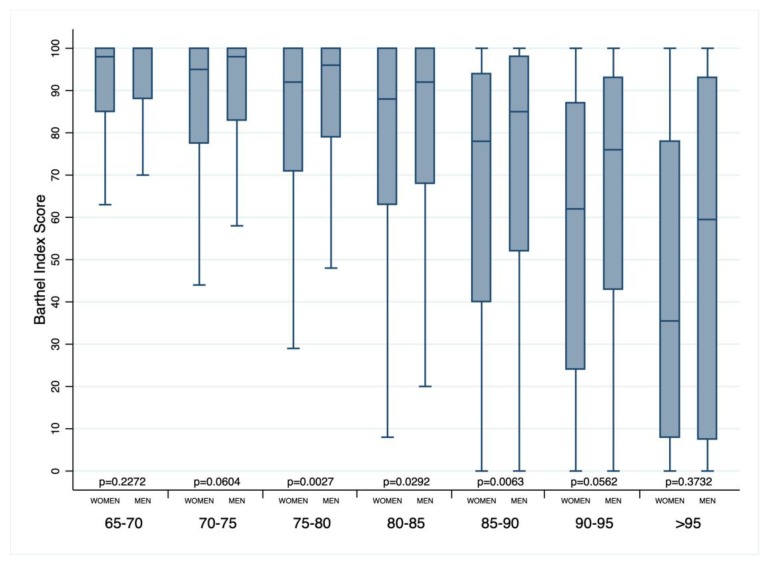
Box-whisker plot of the Barthel index according to sex and age classes.

**Figure 3 jcm-08-00081-f003:**
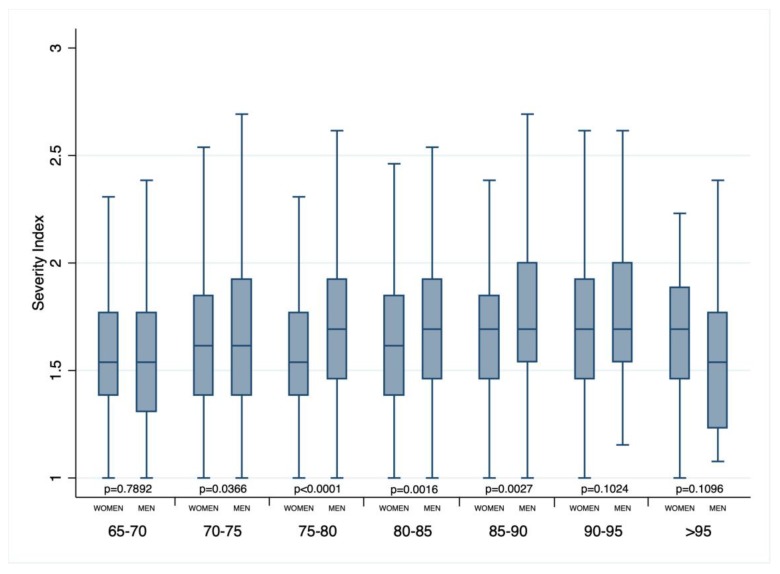
Box-whisker plot of severity index according to sex and age classes.

**Figure 4 jcm-08-00081-f004:**
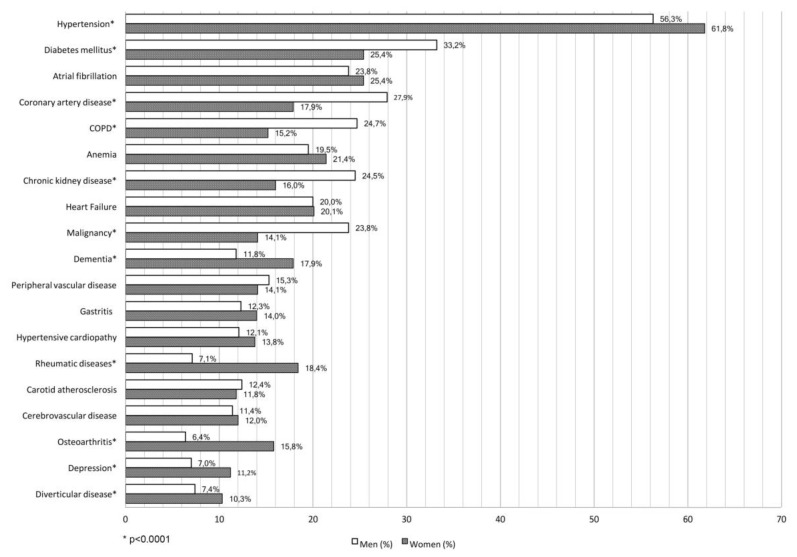
Comparison according to the sex of the most frequent diagnoses (frequency more than 10%) in the RePoSi population (* *p* < 0.001).

**Figure 5 jcm-08-00081-f005:**
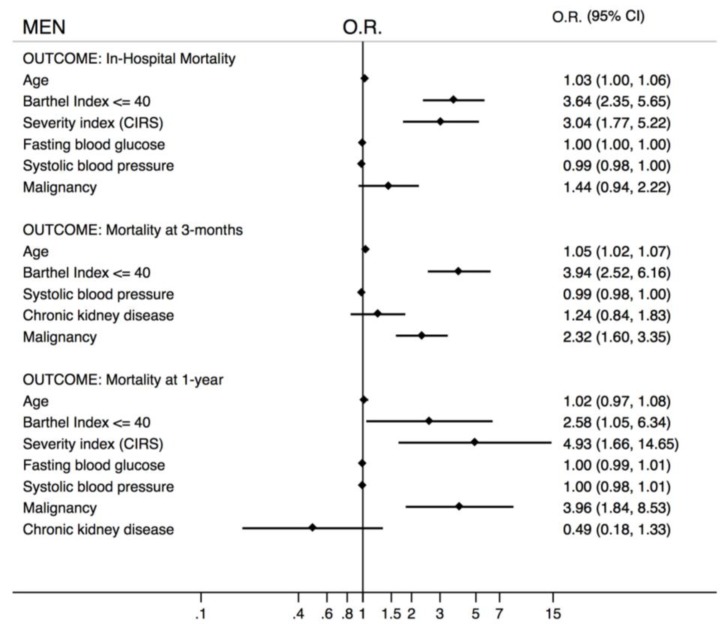
Multivariate analysis according to in-hospital, 3-month, and 1-year mortalities in the male population (OR = odds ratio; 95% CI = 95% confidence interval).

**Figure 6 jcm-08-00081-f006:**
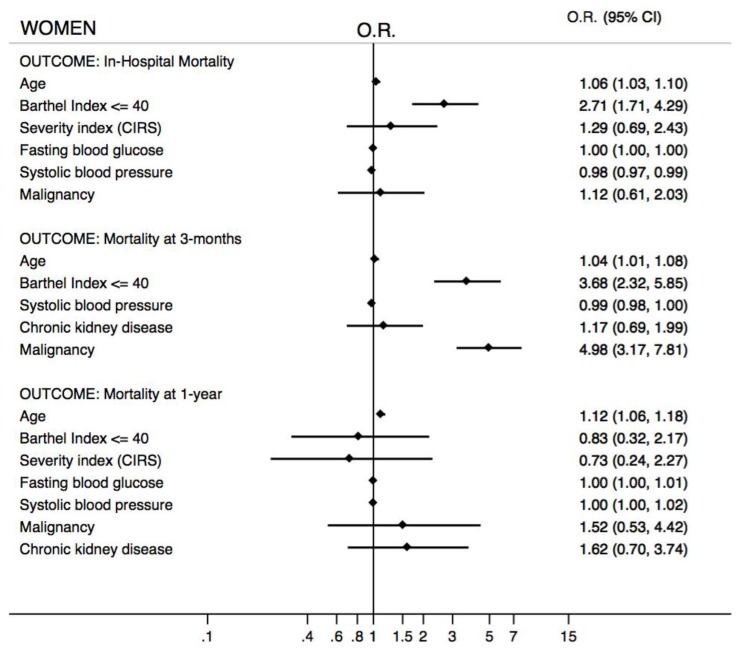
Multivariate analysis according to in-hospital, 3-month, and 1-year mortalities in the female population (OR = odds ratio; 95% CI = 95% confidence interval).

**Table 1 jcm-08-00081-t001:** Clinical characteristics of the RePoSi population at hospital admission.

Variables	Women	Men	*p*
*N* of subjects	2401	2313	/
Age (years) ^a^	80.3 (80–80.7)	78.4 (78.1–78.7)	<0.0001
Systolic blood pressure (mm Hg) ^a^	132.1 (131.2–132.9)	131.7 (130.8–132.5)	0.5840
Diastolic blood pressure (mm Hg) ^a^	73.5 (73.0–73.9)	73.5 (73.1–74.0)	0.7881
eGFR (mL/min) ^a^	58.1 (57.1–59.0)	60.5 (59.5–61.6)	0.0003
Hemoglobin (g/L) ^a^	11.6 (11.5–11.7)	12.1 (12.0–12.2)	<0.0001
Body mass index (kg/m^2^) ^a^	26.1 (25.8–26.3)	25.8 (25.6–25.9)	0.7880
Barthel index (disability) ^a^	74.5 (73.3–75.8)	80.6 (79.5–81.8)	<0.0001
Barthel index ≤ 40 (disability) (%)	17.1	11.8	<0.0001
Geriatric Depression Scale (GDS) ^a^	1.51 (1.45–1.56)	1.27 (1.22–1.33)	<0.0001
Comorbidity index (CIRS) ^a^	2.92 (2.84–2.99)	3.18 (3.10–3.26)	<0.0001
Severity index (CIRS) ^a^	1.64 (1.63–1.65)	1.69 (1.68–1.71)	<0.0001
Length of hospital stay (days) ^a^	11.8 (11.3–12.3)	11.9 (11.3–12.6)	0.2413
In-hospital mortality (%)	4.8	6.1	0.0545
3-month mortality (%)	13.9	19.7	<0.0001
1-year mortality (%)	46.8	58.2	0.0001
Fasting glucose (mmol/L) ^a^	125.6 (123.2–128.0)	129.4 (126.6–132.2)	0.2013
Creatinine (μmol/L) ^a^	1.12 (1.09–1.16)	1.41 (1.37–1.45)	<0.0001
Cholesterol (mmol/L) ^a^	168.3 (166.1–170.4)	150.6 (148.5–152.7)	<0.0001
Cognitive impairment (SBT score) ^a^	9.74 (9.40–10.09)	8.50 (8.17–8.84)	<0.0001
Cognitive impairment (SBT score ≥ 10) (%)	39.0	32.7	<0.0001
Number of drugs at hospital admission ^a^	5.6 (5.5–5.7)	5.9 (5.8–6.0)	0.0022
Number of drugs at hospital discharge ^a^	7.6 (7.5–7.8)	7.7 (7.5–7.9)	0.2524
Number of drugs at follow-up 3-month ^a^	6.5 (6.3–6.6)	6.7 (6.5–6.9)	0.0983
Number of drugs at follow-up 1-year ^a^	6.4 (6.0–6.8)	6.5 (6.1–7.0)	0.5517

^a^ Data are reported as means (95% confidence interval). Abbreviations: BMI = Body Mass Index (underweight BMI < 18.5; optimal weight BMI, 18.5 to 24.9; overweight BMI, 25 to 29.9; obesity I BMI, 30 to 34.9; obesity II BMI, 35 to 39.9; and obesity III BMI ≥ 40), CIRS = cumulative illness rating scale, and SBT = Short Blessed Test (normal cognition SBT < 5; questionable dementia SBT, 5–9; and dementia SBT ≥ 10).

**Table 2 jcm-08-00081-t002:** Multivariate analyses according to in-hospital, 3-month, and 1-year mortalities in all population.

Variables	In-Hospital Mortality		Mortality at 3-Months		Mortality at 1-Year	
	Odds Ratio (95% C.I.)	*p =*	Odds Ratio (95% C.I.)	*p =*	Odds Ratio (95% C.I.)	*p =*
Age	1.05 (1.03–1.07)	<0.0001	1.05 (1.04–1.07)	<0.0001	1.05 (1.03–1.07)	<0.0001
Male sex	1.42 (1.06–1.92)	0.02	1.86 (1.49–2.32)	<0.0001	1.78 (1.35–2.34)	<0.0001
Barthel Index ≤ 40	3.08 (2.24–4.24)	<0.0001	4.19 (3.28–5.37)	<0.0001	3.10 (2.17–4.42)	<0.0001
Severity index (CIRS)	2.01 (1.34–3.03)	0.001	-	-	1.70 (1.12–2.59)	0.013
Fasting blood glucose	1.16 (1.04–1.28)	0.007	-	-	1.10 (0.98–1.22)	0.106
Systolic blood pressure	0.98 (0.98–0.99)	<0.0001	0.99 (0.98–0.99)	<0.0001	0.99 (0.98–0.99)	<0.0001
Chronic kidney disease	-	-	1.15 (0.90–1.47)	0.257	1.05 (0.74–1.49)	0.767
Malignancy	1.35 (0.95–1.90)	0.09	2.48 (1.96–3.14)	<0.0001	4.39 (3.06–6.30)	<0.0001

The model reports only variables with *p* < 0.2; see Statistical Analysis Section.

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
