# Peer review of "Sex-Differences in the Pattern of Comorbidities, Functional Independence, and Mortality in Elderly Inpatients: Evidence from the RePoSI Register"

_jcm, 2019, doi:10.3390/jcm8010081_

Round 1

Reviewer 1 Report

With interest and pleasure I have read the MS Gender-differences in the pattern of comorbidities, 2 functional independence and mortality in elderly 3 inpatients: evidence from the RePoSI register by Corrao et al.

The study protocol and design are known, aims are clear.

I do not understand very well the novelty of the findings. Previous findings of the REPOSI study are already published. This study has, therefore, in my opinion a kind of confirmatory, cumulative character.

Alternatively, the authors should stress, clearly an biologically, differences in findings between men and women.

Minor detail, in modern studies the term seems more appropriate than sex, sinds gender reflects also characteristics others than biology only indicate

Author Response

Thank you for your helpful comments to improve our manuscript:

•    We agree with your statement about the confirmatory purpose of our study. Indeed, it is a validation of our previous work. However, in this analysis, there is a greater statistical consistency through a bigger sample size and newly available data with outcomes at 1 year as well. In this way, our data could be used to provide better gender personalized care to elderly patients. However, according to your suggestion, we stressed biological and genetic differences in findings between men and women. (page 8, lines 173-179; page 9, lines 189-195; page 9, lines 199-205)

Reviewer 2 Report

This study showed gender dimorphism in hospitalized elderly patients and data shown in this article was useful for the gender-personalized-health-care. However, there was several question in analysis of data and discussion.

Causes of hospitalization were not shown in this data. Disease as cause of hospitalization is important factor effecting the prognosis and cognitive function in the patients. And basal Barthel index or cognitive function before hospitalization is also important to understand the prognosis. As average age was significantly different in male and female patients, multivariable analysis should be adopted for all data in which gender difference was added as a factor. It mean gender corrected multivariable data should be shown to discuss the results.

Author Response

Thank you for your helpful comments to improve our manuscript:

•    We agree with your observation about the importance of the causes of hospitalization, but the aim of the study did not include this issue

•    We agree with your observation about Barthel index or cognitive function, but we do not have data before admission because RePoSi register states to describe the prevalence of diseases and treatments (often polytherapy) in hospitalized elderly patients.

•    According to your suggestion, since gender was a variable independently associated with mortality a multivariate analysis of all data in the results section have been included (table 2, page 6, lines 141-142; pages 6-7, 140-147). In this sense, we decided to present and to comment on two gender-specific multivariates since the aim of the study is to highlight gender differences

Round 2

Reviewer 1 Report

I believe that the MS has ameliorated and might be of interest for the readers of the journal

Author Response

Thank you for your positive comment